# Application of the Transition State Theory in the Study of the Osmotic Permeabilities of AQP7, AQP10 and GlpF

**DOI:** 10.3390/membranes15090265

**Published:** 2025-09-02

**Authors:** Ruth Chan, Liao Y. Chen

**Affiliations:** Department of Physics, The University of Texas at San Antonio, San Antonio, TX 78249, USA

**Keywords:** aquaglyceroporin, osmotic permeability, transition state theory, molecular dynamics simulation

## Abstract

Aquaglyceroporins, including human AQP7, AQP10, and *E. coli* GlpF, are known to facilitate movements of glycerol, water, and some other uncharged molecules across the cell membrane. In this study we focused on the transport of water molecules in the absence of glycerol for AQP7, AQP10 and GlpF using the Transition State Theory for the novel application of permeability and kinetics studies. We conducted around 500 ns of in silico simulations of the aquaglyceroporins embedded in lipid bilayer membranes with intracellular-extracellular asymmetries in leaflet lipid compositions. For the water permeability analysis, we computed the transition rate constant with correction for recrossing events where the water molecules do not completely traverse the protein channel from one side of the membrane to the other side. We also studied the hydrogen bond distributions of the single-file waters and channel residues and linear water densities along the pores of the aquaglyceroporins. Interestingly, we found that there was an inverse correlation between the number of single-file water molecules in the channel and osmotic permeability.

## 1. Introduction

Aquaporins (AQPs), a large family of integral membrane proteins, are found in humans and many other living organisms. They may be permeable to water exclusively (orthodox aquaporins), and some are also permeable to small uncharged solutes such as glycerol (aquaglyceroporins). AQPs play a vital role in osmoregulation, which is essential to all life forms. Amongst the aquaglyceroporins, the crystal structure of the glycerol uptake facilitator protein (GlpF) from *Escherichia coli* (*E. coli*) had been long established [1], enabling extensive in vitro and theoretical-computational studies [2,3,4,5,6,7,8,9,10,11,12,13,14,15,16]. Conversely, the high-resolution crystal structures were more recently identified for AQP10 in 2018 [17] and for AQP7 in 2020 [18,19]. Many previous studies on these ‘newer’ aquaglyceroporins, AQP7 [18,19,20,21,22,23,24,25,26] and AQP10 [17,27,28,29], were thus largely in vitro.

Over the years, various computational models have been developed to study, e.g., transport kinetics of carrier and channel proteins at the atomic level and nanosecond timescale. Molecular dynamics simulations help complement the limitations of in vitro studies, thus enabling a greater overall understanding of these proteins essential to the life of the cells within the organism. Zhu et al. introduced the computational method utilizing the creation of a hydrostatic pressure gradient under periodic boundary conditions to observe the net water flow through the channels [30]. With a hydrostatic pressure difference established by subjecting the water molecules to a constant force, along with constraints and counterforces on the membrane/protein and elimination of the force on the water molecules inside the channel, the single-channel osmotic permeability could be calculated from the retrieval of the single-channel hydraulic permeability. At the time, computational power was limited with respect to today, and thus large pressure differences were required to yield results from shorter simulations. A later study related the concentration differences in the impermeant solute, ∆C_S_ (mol/cm^3^) and water flux, *j_W_* (mol/s), with the water chemical potential, ∆µ (termed the collective diffusion model). This allowed the study of water movement in channels under equilibrium and non-equilibrium conditions by utilizing the development of a random walk approximation for the water molecules in the channel. With this model, calculating the osmotic permeability was equated to calculating the diffusion coefficient [31]. Subsequently, Wambo et al. proposed the method of computing water flux induced by a small NaCl osmolyte concentration gradient (in the mM range) across the membrane, which yielded results with a greater level of accuracy. To increase the signal-to-noise ratio, a large model system was used with four tetramers embedded in a large patch of lipid bilayer representing the cellular membrane [3].

In 2020, our lab utilized the Transition State Theory (TST) for a novel application in the computation of osmotic permeabilities of AQP1 and AQP3 [32]. Developed in the 1930s by Eyring et al., the TST provides understanding of thermally activated processes through changes in the free-energy landscape along the reaction pathway. The TST method has been traditionally applied, for example, in the study of chemical reactions and the unfolding of proteins and RNA structures. Expanding upon the TST fundamental dynamical assumption (or no-recrossing assumption), we introduced the correction factor κ to exclude events of water molecules recrossing the dividing plane. Considering the rugged potential energy landscape of the aquaporin channel, we set the dividing plane *z* = *z*_0_ as the most constricted point in the aquaporin channel and evaluated κ by following every event trajectory of a molecule crossing the dividing plane from the intracellular (IC) side to the extracellular (EC) side until it either recrosses the plane back to the IC side or moves out of the channel into the EC bulk (or vice versa from EC to IC). The validity of our method was supported by the in vitro results from the well-established protocol of cell swelling/shrinkage studies using stopped-flow experiments on erythrocytes [32].

In this study, we further applied the TST method specifically for the study of aquaglyceroporins AQP7, AQP10, and GlpF for the computation of osmotic permeabilities at 5 °C and 25 °C. We excluded glycerol from the simulations, as it has been previously shown that aquaglyceroporins have a high affinity for glycerol, which negatively affects osmotic permeability [5,18,33,34,35,36]. As will be detailed later in this paper, we found that the water permeability of AQP10 was the highest, followed by GlpF, and then AQP7. Interestingly, our results agreed with Horner et al., who found a direct correlation between the number of single-file water molecules in the channel and the permeability of water through the calculation of single-channel osmotic permeability, *p_f_*, with great accuracy from in vitro studies [4]. Previous experimental and computational results on GlpF also concurred with our findings [3,4], which confirmed the validity of our method. To further study the basis of the differences in permeability, we also examined the linear density of water in the channels and the differences in the selectivity filter (sf) of these aquaglyceroporins.

## 2. Methods

### 2.1. Model System Setup and Simulation Parameters

CHARMM-GUI was used to set up the all-atom models of the GlpF, AQP7 and AQP10 tetramers, whose coordinates were taken from PDB ID 1FX8 [1], 6QZI [18] and 6F7H [17], respectively. Each individual tetramer was embedded in a 117 Å × 117 Å patch of model membrane with the lipid compositions of 20% cholesterol (CHL1), 30% phosphocholine (POPC), and 49% phosphatidylethanolamine (POPE) for AQP7; 68% POPE, 27% phosphoglycerol (POPG), and 5% phosphocardiolipin (PMCL2) for GlpF; and 20% CHL1 and 80% POPE for AQP10 (Figure 1, Figure 2 and Figure 3). The positioning of the tetramers was determined by matching the hydrophobic side surface with the lipid tails and aligning the channel axes perpendicular to the membrane. The AQP-membrane complex was sandwiched between two layers of TIP3P waters, each of which was approximately 30 Å thick. The system was then neutralized and salinated with Na^+^ and Cl^−^ ions to a salt concentration of 150 mM. We employed NAMD 2.13 [37] as the molecular dynamics (MD) engine and used CHARMM36 parameters [38,39,40] for inter- and intra-molecular interactions. After the initial equilibration steps, we fully equilibrated the system by running unbiased MD for 200 to 500 ns with constant pressure at 1.0 bar (Nose–Hoover barostat) and constant temperature at 278.15 K and 298.15 K (Langevin thermostat), respectively. The Langevin damping coefficient was chosen to be 1/ps. The period boundary conditions were applied to all three dimensions. The particle mesh Ewald (PME) was used for the long-range electrostatic interactions (grid level: 128 × 128 × 128). The time step was 2.0 fs. The cut-off for long-range interactions was set to 10 Å with a switching distance of 9 Å.

### 2.2. Transition State Theory for Calculation of Water Permeability

The permeability of water was calculated using the transition state theory with correction for recrossing events as detailed in Ref. [32]. Here we repeat the prescribed steps: An osmotic gradient is generated by an imbalance in impermeable osmolyte concentrations between two sides of a membrane. In the absence of hydraulic pressure difference, the intracellular (IC) to extracellular (EC) transition rate constant of water *k_ItoE_* is related to the EC-to-IC rate constant *k_EtoI_* as follows:(1)$$k_{ItoE}=k_{EtoI}exp\left[(c_{e}-c_{i})\nu_{W}\right].$$

The EC and IC concentrations of the impermeable solutes are denoted by *c_e_* and *c_i_*, respectively, and *ν_W_* is the molar volume of water. Under a hyperosmotic condition, $c_{e}-\;c_{i}>0$; therefore, $k_{ItoE}>\;k_{EtoI}$, and the outward chemical potential gradient induces a net efflux of water. The corresponding transition rate (transitions per unit time facilitated by one AQP channel) is(2)$$r=k_{ItoE}\left(1/\nu_{W}-c_{i}\right)-k_{EtoI}\left(1/\nu_{W}-c_{e}\right).$$

Note that $1/\nu_{W}$ is the concentration of water molecules in the absence of solutes. The presence of solutes reduces the water concentration. Furthermore, the validity of these formulas is limited to the dilute solution regime, namely, $\left(c_{e}-\;c_{i}\right)\nu_{W}\ll 1$, which is quantitatively accurate for osmolyte concentration in the sub-molar range. In this regime, the linear expansion in terms of $\left(c_{e}-\;c_{i}\right)$ is valid, which leads to the transition rate(3)$$r=2k_{0}\left(c_{e}-c_{e}\right).$$

Therefore, the water flux through a single AQP channel (the volume of water flowing through a channel per unit time),(4)$$J=r\nu_{W}/N_{A}=2k_{0}(c_{e}-c_{i})\nu_{W}/N_{A}.$$

Correspondingly, the single-channel permeability(5)$$p_{f}=J/\left(c_{e}-c_{i}\right)=2k_{0}\nu_{W}/N_{A}.$$

Here, *k*_0_ is the rate constant $k_{0}=\;k_{EtoI}=\;k_{ItoE}$ at equilibrium ${(c}_{e}=\;c_{i})$ and *N_A_* is the Avogadro number.

With the dividing plane between the IC and EC sides of z = 0, only events of water molecules crossing the dividing plane in the positive z-direction and continuing onto the EC side (for IC-to-EC transitions, and vice versa for EC-to-IC transitions) without recrossing the dividing plane are considered. Therefore, in the application of the transition state theory (TST), this gives(6)$$k_{0}=\kappa <\nu_{z}\delta \left(z-z_{0}\right){>}_{v_{Z}>0},$$
where κ is the correction factor and *z* = *z*_0_ = 0 is the dividing plane. The brackets indicate the equilibrium statistical average of the velocity along the z-direction (or equivalently the negative z-direction for the EC-to-IC rate constant). Carrying out the equilibrium statistical average, we have(7)$$k_{0}=\kappa \sqrt{RT/2\pi m_{W}n(z_{0})},$$
with *m_W_* being the molar mass of water. The linear density of water at the dividing plane *n*(*z*_0_) can be readily evaluated with equilibrium sampling, which was computed from scripts modified from Wang et al. [41]. The correction factor *κ* is equal to the number of successful transport events (ending up in the EC bulk at (*z* > 15) or IC bulk (*z* < −15)) divided by the total number of events of crossing the dividing plane.

We also calculated the Arrhenius activation energy for the systems from the permeability values at 5 °C and 25 °C for the analysis of the effect of temperature on the permeability of the protein channels. The Arrhenius activation energy, *E*_a,_ was computed using the formula(8)$$E_{a}=\frac{RT_{1}\ln\left(\frac{p_{1}}{p_{2}}\right)}{\frac{T_{1}}{T_{2}}-1},$$
where R is the gas constant, T_1_ and T_2_ are the two temperatures in Kelvin, and *p*_1_ and *p*_2_ are the two corresponding permeabilities.

## 3. Results and Discussion

### 3.1. Single-Channel Osmotic Permeability, p_f_, of AQP7, GlpF and AQP10

Following equilibration of the systems, as confirmed by RMSD analysis (Appendix A), we ran the equilibrium MD for an additional 100 ns for each system, counting the permeation events for the analysis of permeabilities. Using the TST method, the total number of permeation attempts by water molecules from both the EC and IC sides were computed over the equilibrated simulation length for all AQPs at 5 °C and 25 °C. The total number of successful events was also calculated (multiplied by 50 to allow the graph to be on the same scale) (left columns of Figure 4 and Figure 5), with the correction for recrossing events. The success ratios (%), i.e., the κ values were obtained by dividing the total number of successful events of permeation by the total number of attempts. It can be seen at both temperatures that for GlpF and AQP10 there was a similar net transport of water molecules from one side of the membrane to the other, which was greater compared to AQP7 (right columns of Figure 4 and Figure 5). As the κ values remained steady throughout the simulation duration, it can be inferred that the amount of water molecules successfully traversing the channel over time could be due to the inherent characteristics of the channels.

Permeability values of *p_f_* were illustrated as a function of time in Figure 4 and Figure 5, with the average values and error margins listed in Table 1. Upon examination, there is a consistent trend at both temperatures, with the highest permeability seen for AQP10 (17.929 ± 0.028 at 25 °C; 11.395 ± 0.028 at 5 °C), followed by GlpF (15.304 ± 0.098 at 25 °C; 9.657 ± 0.033 at 5 °C), and then AQP7 (6.839 ± 0.047 at 25 °C; 4.664 ± 0.019 at 5 °C). Also, with the increase in temperature from 5 °C to 25 °C, as expected, osmotic permeability increased for all aquaglyceroporins due to the rise in thermal fluctuations. Existing osmotic permeability in vitro data for GlpF, AQP7 and AQP10 vary considerably over several orders of magnitude from 10^−13^ cm^3^/s per channel. This is mainly due to the difficulty in determining the density of water channels from the various cellular/reconstituted systems to calculate single-channel transport rates from membrane permeability experimental assays. However, in a recent study by Horner et al., high-precision GlpF permeability values were found through rigorous experimental methods to measure the abundance of reconstituted aquaporins [4]. The experimentally derived permeability of GlpF to water at 5 °C (11.7 × 10^−13^ cm^3^/s) [4] was in good agreement with our results (9.66 × 10^−13^ cm^3^/s), and similarly for the estimated value at 25 °C calculated by Horner et al. from the known activation energy (19.0 × 10^−13^ cm^3^/s) in comparison to our values (15.3 × 10^−13^ cm^3^/s).

### 3.2. Effect of Temperature on Osmotic Permeability

The Arrhenius activation barrier or energy for water permeation by all three aquaglyceroporins was remarkably similar at approximately 3.7–3.8 kcal/mol for GlpF and AQP10 and 3.2 kcal/mol for AQP7 for the simulations that were performed in the absence of glycerol (Table 1). This is in line with experimental values for AQP7 [22] and GlpF [4] and, in general, indicates water movement through aqueous pores [42]. With glycerol present in the experiment, it has been reported that for GlpF the *E_a_* value is 7 kcal/mol [12], showing that bound glycerol in the channel occludes other water or glycerol molecules from traversing it.

We also looked at the RMSF values of the NPA motifs and the aromatic/arginine (ar/R) selectivity filter residues, respectively, and found that they were not significantly different between the studied aquaglyceroporins or between the two temperatures (Appendix A). The values indicated that the average fluctuations of the amino acids in these regions (including the backbone and side chain) remained relatively stable throughout the simulation. This corroborated the findings from the E_a_ values that the aquaglyceroporin water permeabilities were less dependent on temperature changes.

### 3.3. Correlation Between Osmotic Permeability and Water–Protein Hydrogen Bonds

Figure 6 depicts the probability histograms of the hydrogen bonds between single-file water molecules in the channels (in blue) and between the water molecules and pore-lining residues for all three aquaglyceroporins (in orange) at 25 °C. These values were averaged across the monomers (values for individual monomers are in Appendix A for 25 °C, and Appendix A for 5 °C). Also shown are the MD snapshots of single-file waters in the respective channels. The approximate number of water–protein hydrogen bonds was ~6.5 for AQP7, followed by ~5.5 for GlpF, and ~5.0 for AQP10 at both 25 °C (Figure 6) and 5 °C (Appendix A). For all three aquaglyceroporins, the number of water–water hydrogen bonds was consistently lower than the water–protein hydrogen bonds, i.e., ~4.5 for AQP7, followed by ~4 for GlpF and ~3.5 for AQP10. Interestingly, the number of hydrogen bonds between the single-file waters and channel residues inversely correlated with the permeability values of AQP7, GlpF and AQP10, i.e., increasing hydrogen bonds (or single-file waters) decreases aquaglyceroporin osmotic permeability. For AQP10, the reduced interaction of water molecules with the channel residues coupled with the widened channel radius may have contributed to the increased permeability. Additionally, amongst the aquaglyceroporins studied, AQP10 has the shortest region of single-file waters, with most of the water molecules in the channel in a more compacted and bulk-like state. Single-file waters act as a rate-limiting step, preventing water molecules from overtaking each other and only permitting a single-direction water flow at any instance. Similarly, other studies have found that water permeability is influenced by the interactions between water and the pore-lining residues [43,44], and notably Horner et al. found that *p_f_* increased exponentially with a decreasing number of hydrogen bond donating or accepting residues in the channel wall [4,45].

### 3.4. Effect of Channel Structure on Osmotic Permeability

To visualize the channel shape and dimensions in terms of how many water molecules can compact within a single channel, we calculated the water linear density values along and normal to the z-axis (Figure 7). This provided a representative depiction of the average channel capacity for water molecules throughout the simulation length upon equilibration. Higher density values indicate a smaller distance between the molecules (i.e., greater water packing) and a greater degree of disorder. Conversely, lower density values indicate that the water molecules are spaced further apart and are thus closer to the single-file configuration. As expected, the z-axis regions with low density values (≤2.5 molecule/nm^3^) were the shortest for AQP10 (~10 Å), followed by GlpF (~12 Å), and then AQP7 (~14 Å) (Figure 7). At both the intracellular and extracellular openings, the linear density increases at a rapid rate, eventually plateauing at the ends of the channels in contact with the bulk water region. Interestingly, for AQP10 there was a significant reduction in linear densities at 25 °C compared to 5 °C, which was not observed in AQP7 or GlpF. This corroborates with the greater difference in permeability values for AQP10 at the two temperatures compared with the other aquaglyceroporins. Moreover, a constriction in the channel opening at the intracellular side can be clearly seen for AQP10 from the linear density graphs, as was also found by Gotfryd et al. [17].

To further investigate the possible factors influencing permeability, we compared the selectivity filter (sf) regions of the three aquaglyceroporins, focusing on AQP7 and AQP10 in comparison with GlpF. Figure 8a depicts the sf of AQP10, which was considerably wider compared to AQP7 (sf residues indicated in yellow). The relatively open configuration of the AQP10 sf allows multiple water molecules to pass through simultaneously. Conversely, the AQP7 sf was significantly narrower and only allowed passage of water molecules from the EC side in single file. Upon comparison of the AQP7 and AQP10 sf residues with GlpF, there were some significant differences in the configuration (Figure 8b). GlpF residues that contributed to the restriction site are W48, F200 and R206, whereas G191 is typically omitted from sf analysis as it is located further away from the channel lining. Similarly, G202 from AQP10 is also situated away from the channel pore. AQP7, however, was notably different, with G222 directly bonded to Y223 and thus possibly contributing to the decreased osmotic permeability due to the steric hindrance. The G222 residue of AQP7 on the EC side of the sf was almost parallel to the direction of water flow, causing a narrowing of the channel opening near the pore entrance. A closer examination of the superimposition of the sf residues of the aquaglyceroporins clearly showed that the AQP10 sf was significantly wider, as it lacked the characteristic aromatic residues (which restrict the molecules passing through) as found in GlpF and AQP7. Instead, for AQP10, these residues were replaced by smaller hydrophobic amino acids G62 and I211 (Figure 8b). Furthermore, we conducted HOLE [46] analyses of the three aquaglyceroporins. The graphical illustrations of the channels are shown in Appendix A. The channel radii are shown in Figure 9, which shows that, in the sf region (7<z<11), the AQP10 channel is approximately twice as wide as the AQP7/GlpF channel.

## 4. Conclusions

Through MD simulations using the established parameters, we measured the osmotic permeabilities of AQP7, AQP10 and GlpF with the novel application of the TST method. Our results were in agreement with recent experimental studies. From the analysis of the number of hydrogen bonds, linear densities and the selectivity filter, we obtained further insights into the probable factors contributing to the differences in permeabilities between these aquaglyceroporins. Notably, we found that there was an inverse correlation between the number of single-file water molecules in the channel and osmotic permeability. Finally, the osmotic permeabilities of aquaglyceroporins ($\sim 10\times 10^{-13}\;{\mathrm{cm}}^{3}/s$) are significantly higher than that of a water-specific aquaporin (e.g., $\sim 4\times 10^{-13}\;{\mathrm{cm}}^{3}/s$ [32] for AQP1), which indicates that aquaglyceroporins, especially GlpF, are good candidates for water filtration applications.

## Figures and Tables

**Figure 1 membranes-15-00265-f001:**
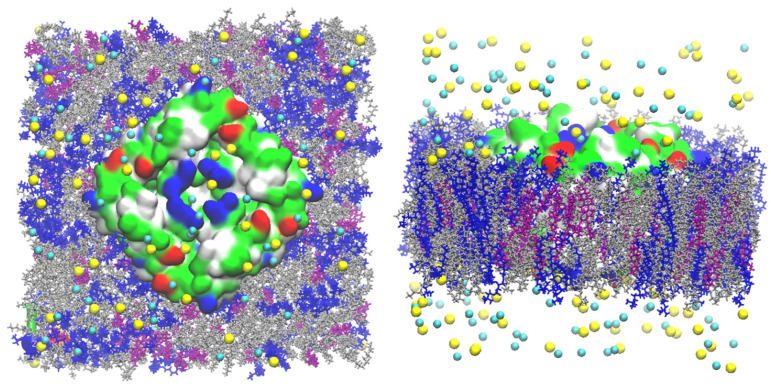
AQP7 model system. In the **left panel** is the top view from the EC side, and in the **right panel** is the side view. The protein tetramer is represented by surfaces colored by residue types (hydrophilic, green; hydrophobic, white; positively charged, blue; negatively charged, red). The POPE lipids are indicated in silver, CHL1 in purple, and POPC in blue. The Na^+^ ions and Cl^−^ anions are shown as yellow and cyan spheres, respectively. Water molecules are not shown for clearer views of all other constituents of the system.

**Figure 2 membranes-15-00265-f002:**
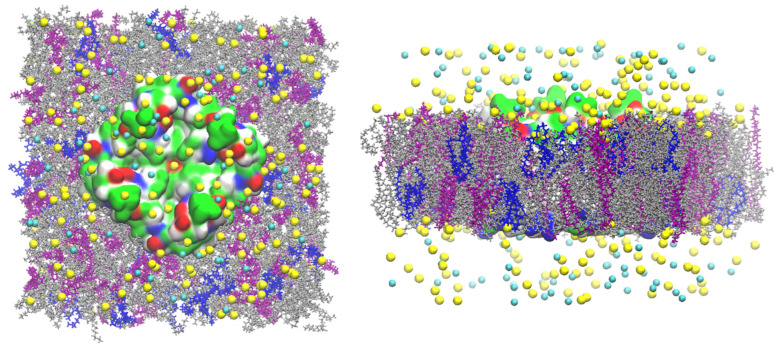
GlpF model system. In the **left panel** is the top view from the EC side, and in the **right panel** is the side view. The protein tetramer is represented by surfaces colored by residue types (hydrophilic, green; hydrophobic, white; positively charged, blue; negatively charged, red). The POPE lipids are indicated in silver, PMCL2 in blue, and POPG in purple. The Na^+^ ions and Cl^−^ anions are shown as yellow and cyan spheres, respectively. Water molecules are not shown for clearer views of all other constituents of the system.

**Figure 3 membranes-15-00265-f003:**
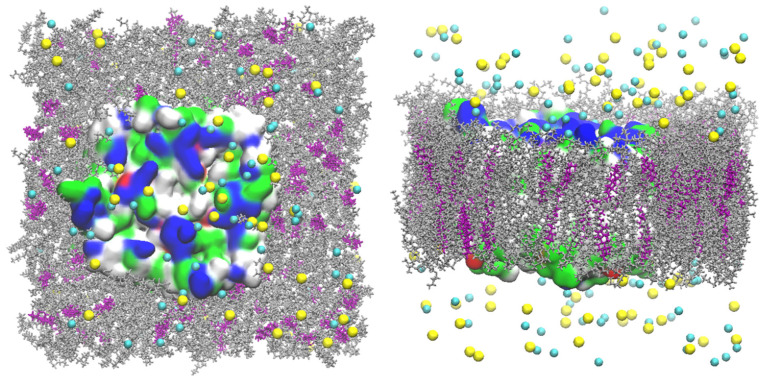
AQP10 model system. In the **left panel** is the top view from the EC side, and in the **right panel** is the side view. The protein tetramer is represented by surfaces colored by residue types (hydrophilic, green; hydrophobic, white; positively charged, blue; negatively charged, red). The POPE lipids are indicated in silver, CHL1 in purple, and POPC in blue. The Na^+^ ions and Cl^−^ anions are shown as yellow and cyan spheres, respectively. Water molecules are not shown for clearer views of all other constituents of the system.

**Figure 4 membranes-15-00265-f004:**
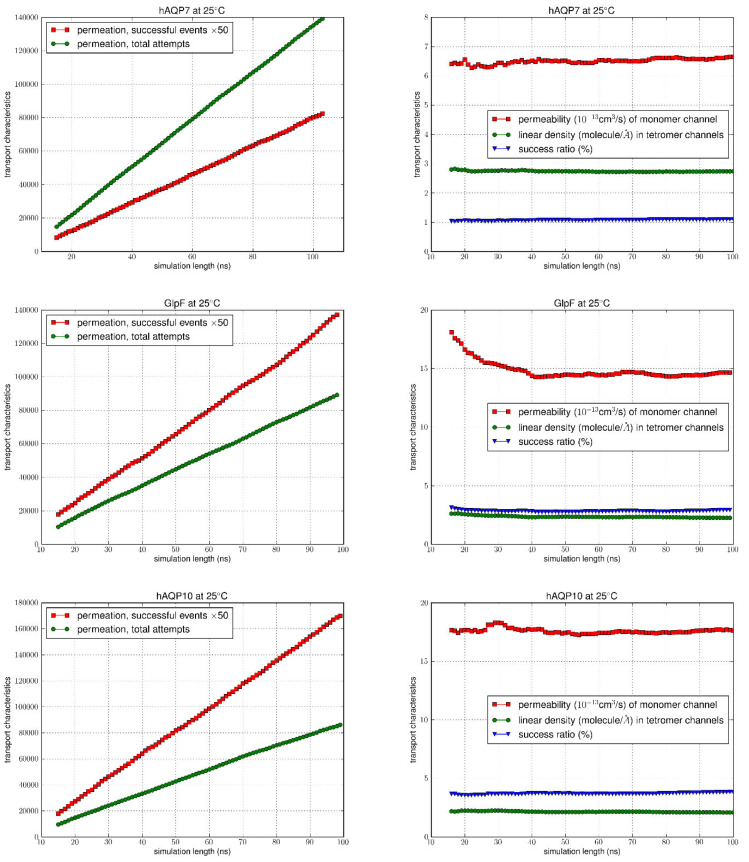
MD simulation results at 25 °C. In the left columns, the number of successful (×50) was compared to the total number of permeation attempts. The vertical axis shows the number of successful events and the total number of attempts during the time interval (0, t), while the horizontal axis is the simulation time t. Columns on the right indicate the permeability values, linear density and success ratio over the simulation length (0, t) for 15 ns < t < 100 ns.

**Figure 5 membranes-15-00265-f005:**
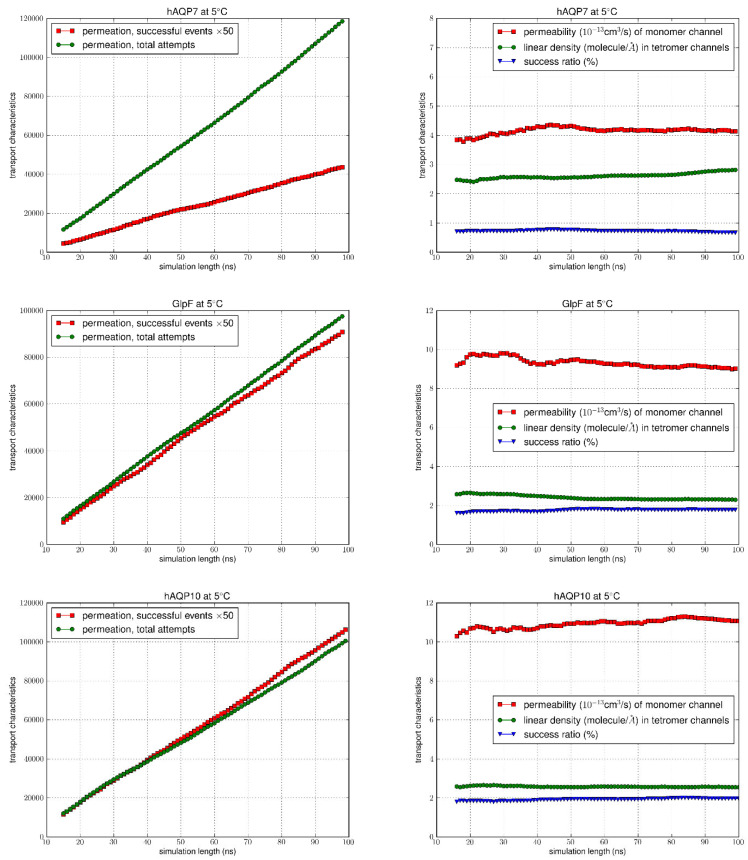
MD simulation results at 5 °C. In the left columns, the number of successful events (×50) was compared to the total permeation attempts. The vertical axis shows the number of successful events and the total number of attempts during the time interval (0, t), while the horizontal axis is the simulation time t. Columns on the right indicate the permeability values, linear density and success ratio over the simulation length (0, t) for 15 ns < t < 100 ns.

**Figure 6 membranes-15-00265-f006:**
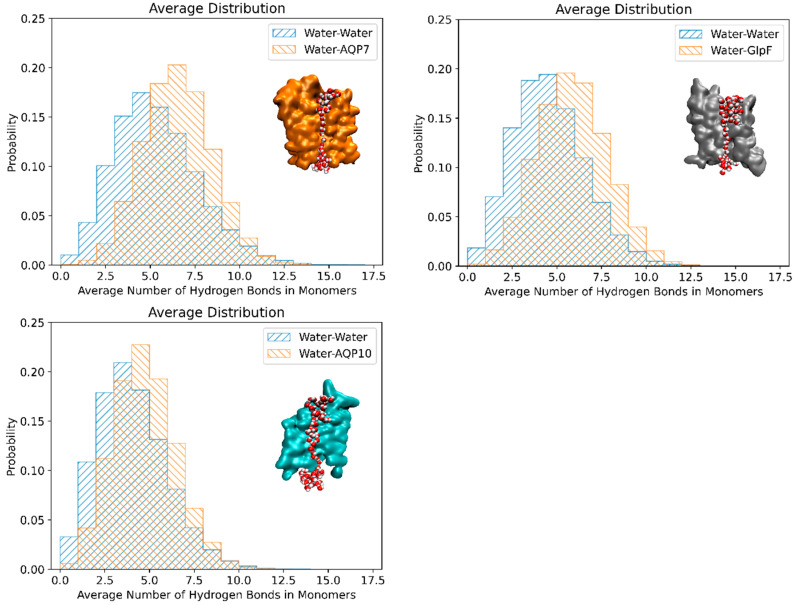
The number of hydrogen bonds formed (i) amongst the single-file waters in the channel (in blue) and (ii) between the single-file waters and pore-lining residues (in orange) of AQP7, GlpF and AQP10, respectively, at 25 °C. The results shown were averaged across all the monomers in the tetramer and throughout the equilibrated simulation duration.

**Figure 7 membranes-15-00265-f007:**
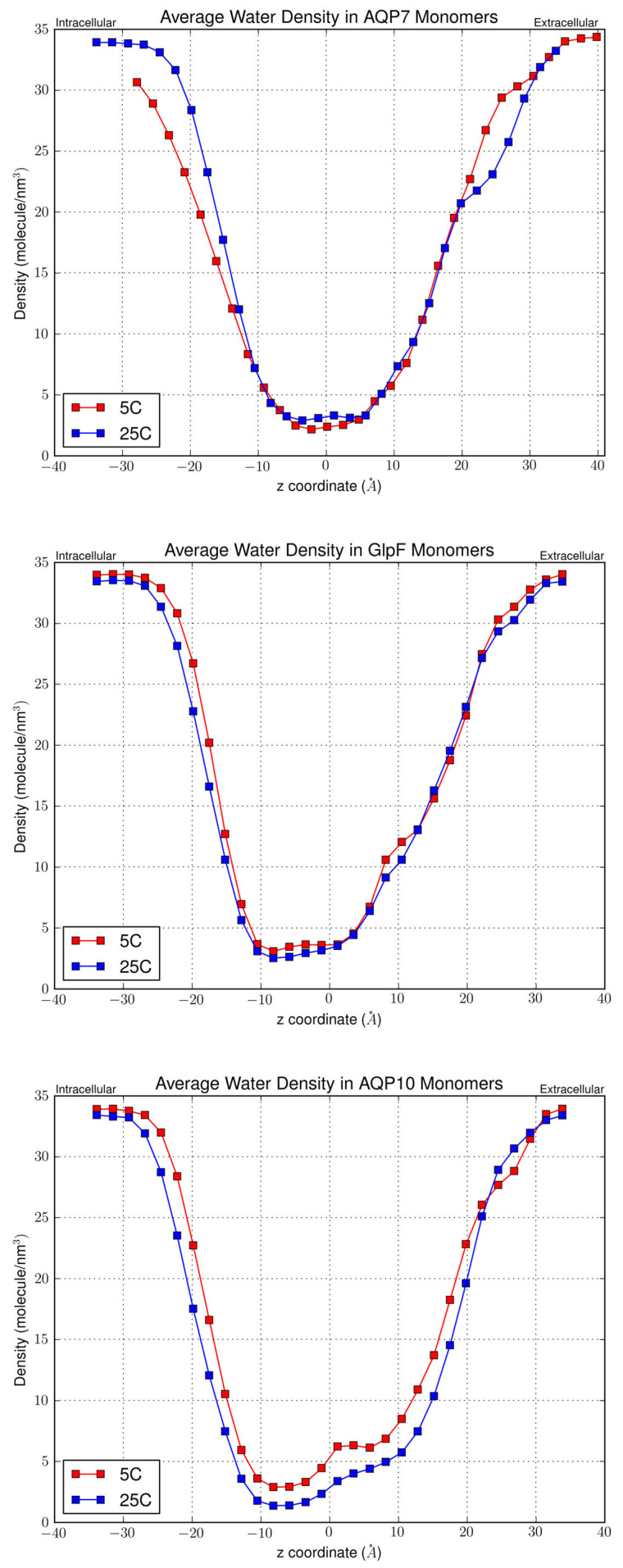
Linear density of water molecules in AQP7, GlpF and AQP10 channels, respectively, at 25 °C and 5 °C. Figures were averaged over the four monomers of each tetramer.

**Figure 8 membranes-15-00265-f008:**
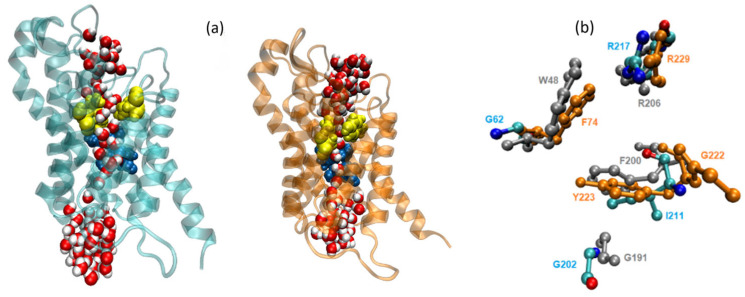
Selectivity filter (sf) and NPA motifs. (**a**) The monomer AQP10 (cyan) is depicted with the sf in yellow and the NPA motif in blue. The AQP7 monomer (orange) is included for comparison. (**b**) Sf residues; AQP10 is shown in cyan, AQP7 in orange, and GlpF in gray.

**Figure 9 membranes-15-00265-f009:**
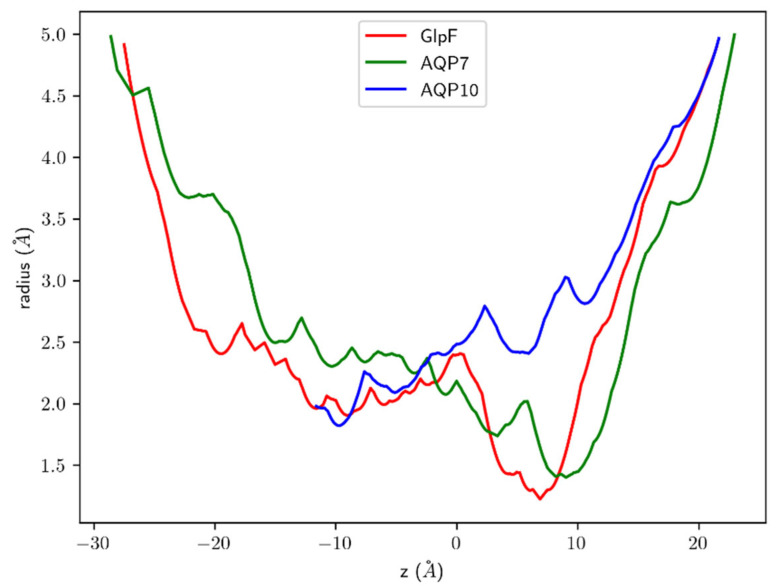
Channel radii of the three aquaglyceroporins after 100 ns of equilibrium MD run. The channels are aligned along the z-axis. The sf regions are located at (7<z<11). For
z<−13, the AQP10 channel’s radius was not computed, as its pore-forming residues fluctuate too much to allow reasonable accuracy.

**Table 1 membranes-15-00265-t001:** Single-channel permeabilities *p_f_* (10^−13^ cm^3^/s) and Arrhenius activation energy, *E*_a_ (kcal/mol) of GlpF, AQP7 and AQP10. (The experimentally measured values and the source references are put inside the parentheses).

AQP	*p_f_* at 5 °C	*p_f_* at 25 °C	*E* _a_
AQP7	4.664 ± 0.019	6.839 ± 0.047	3.154 (2.1 [22])
GlpF	9.657 ± 0.033 (11.7 [4])	15.304 ± 0.098 (19.0 [4])	3.794 (4.0 [4])
AQP10	11.395 ± 0.028	17.929 ± 0.028	3.734

## Data Availability

The dataset (parameters, coordinates, scripts, etc.) to replicate this study will be available upon reasonable requests.

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
