# Peer review of "Application of the Transition State Theory in the Study of the Osmotic Permeabilities of AQP7, AQP10 and GlpF"

_membranes, 2025, doi:10.3390/membranes15090265_

Round 1

Reviewer 1 Report

Comments and Suggestions for Authors

The followings points need to be considered and rectified

Table 1: wherever authors compare or mention that “similar to experimental values” for Ea or Pf etc. please add the corresponding compared value into the table and cite accordingly

Fig4 and 5- Define what “transport characteristic” is?

Fig4 and 5- Clarify if the values in plots represent permeation events within a time window or accumulation for each ns of simulation, clarify this and mention in methods or figure legend

In Methods mention, equations are part of sentence for clarity the equations and notations can be presented better way in individual lines. Also, how k0 etc. are calculated from MD permeation data need to be clearly stated.

Fig 2 and Fig3 can be merged, and Figure resolution should certainly be improved as many labels are blurry

Line 229- ‘Diagrams’ is inaccurate replace with “MD snapshots” or other appropriate word

Line 274-276- Authors should quantify and highlight the difference in SF region via calculating changes in SASA or distance from MD or at least from crystal structure. Use tools like HOLE or MDTAP

Line 286-290- “A closer examination of the superimposition of the sf residues of the aquaglyceroporins clearly showed that the AQP10 sf was significantly wider, as it lacked the characteristic aromatic residues 288(which restricts the molecules passing through) as found in GlpF and AQP7.” – please quantify

In conclusion or in introduction highlight why these simulations that is aquaglyceroporin without glycerol is needed apart from mechanistic value

Author Response

Comment 1:

Table 1: wherever authors compare or mention that “similar to experimental values” for Ea or Pf etc. please add the corresponding compared value into the table and cite accordingly

Response 1:

Available experimental values and their sources are inserted in Table 1.

Comment 2:

Fig4 and 5- Define what “transport characteristic” is?

Response 2:

Now spelled out in the figure captions.

Comment 3:

Fig4 and 5- Clarify if the values in plots represent permeation events within a time window or accumulation for each ns of simulation, clarify this and mention in methods or figure legend

Response 3:

Done.

Comment 4:

In Methods mention, equations are part of sentence for clarity the equations and notations can be presented better way in individual lines. Also, how k0 etc. are calculated from MD permeation data need to be clearly stated.

Response 4:

Done as suggested.

Comment 5:

Fig 2 and Fig3 can be merged, and Figure resolution should certainly be improved as many labels are blurry

Response 5:

Figs. 1 to 3 are now replaced with high-resolution graphics. It seems to us clearer to keep Fig. 2 and Fig. 3 unmerged.

Comment 6:

Line 229- ‘Diagrams’ is inaccurate replace with “MD snapshots” or other appropriate word

Response 6:

Done.

Comment 7:

Line 274-276- Authors should quantify and highlight the difference in SF region via calculating changes in SASA or distance from MD or at least from crystal structure. Use tools like HOLE or MDTAP

Response 7:

HOLE analyses are now added to Fig. 9 and SI (Figs. S10 to S12).

Comment 8:

Line 286-290- “A closer examination of the superimposition of the sf residues of the aquaglyceroporins clearly showed that the AQP10 sf was significantly wider, as it lacked the characteristic aromatic residues 288(which restricts the molecules passing through) as found in GlpF and AQP7.” – please quantify

Response 8:

Done.

Comment 9:

In conclusion or in introduction highlight why these simulations that is aquaglyceroporin without glycerol is needed apart from mechanistic value

Response 9:

Done.

Reviewer 2 Report

Comments and Suggestions for Authors

The manuscript is well organized and clearly written. It investigates the water transport properties of three aquaglyceroporins (AQP7, AQP10, and GlpF) using transition state theory in combination with molecular dynamics simulations in lipid bilayer membranes, excluding glycerol. The study analyzes permeability rates, HB patterns, and linear water densities, providing insights into the correlation between single-file water occupancy and osmotic permeability. The Supp. materials file is missing, and I require it to verify the reported contents before final decision. I recommend acceptance after minor revisions, pending submission of the Supp. file.

Author Response

Comments and Suggestions for Authors

The manuscript is well organized and clearly written. It investigates the water transport properties of three aquaglyceroporins (AQP7, AQP10, and GlpF) using transition state theory in combination with molecular dynamics simulations in lipid bilayer membranes, excluding glycerol. The study analyzes permeability rates, HB patterns, and linear water densities, providing insights into the correlation between single-file water occupancy and osmotic permeability. The Supp. materials file is missing, and I require it to verify the reported contents before final decision. I recommend acceptance after minor revisions, pending submission of the Supp. file.

Response:

We improved Figs. 1 to 3, added Fig. 9 in the main text, and added Figs. S10 to S12 in the Supplemental Information.